# Centromeric KIR AA Individuals Harbor Particular KIR Alleles Conferring Beneficial NK Cell Features with Implications in Haplo-Identical Hematopoietic Stem Cell Transplantation

**DOI:** 10.3390/cancers12123595

**Published:** 2020-12-01

**Authors:** Léa Dubreuil, Bercelin Maniangou, Patrice Chevallier, Agnès Quéméner, Nolwenn Legrand, Marie C. Béné, Catherine Willem, Gaëlle David, Mehdi Alizadeh, Dhon Roméo Makanga, Anne Cesbron, Ketevan Gendzekhadze, Katia Gagne, Christelle Retière

**Affiliations:** 1Etablissement Français du Sang, F-44011 Nantes, France; lea.dubreuil@efs.sante.fr (L.D.); bercelin.maniangouzonzeka@chu-nantes.fr (B.M.); nolwenn.legrand@efs.sante.fr (N.L.); catherine.willem@efs.sante.fr (C.W.); gaelle.david@efs.sante.fr (G.D.); dhon.makanga@efs.sante.fr (D.R.M.); anne.cesbron@efs.sante.fr (A.C.); 2INSERM U1232 CNRS, CRCINA, Université de Nantes, F-44000 Nantes, France; patrice.chevallier@chu-nantes.fr (P.C.); Agnes.Quemener@univ-nantes.fr (A.Q.); mariechristine.bene@chu-nantes.fr (M.C.B.); 3LabEx IGO “Immunotherapy, Graft, Oncology”, F-44000 Nantes, France; 4Hematology Clinic, CHU, F-44000 Nantes, France; 5Hematology Biology, Nantes University Hospital, F-44000 Nantes, France; 6Etablissement Français du Sang, F-35016 Rennes, France; mehdi.alizadeh@efs.sante.fr; 7LabEx Transplantex, Université de Strasbourg, F-67000 Strasbourg, France; 8Department of Hematology and HCT, HLA laboratory, City of Hope Medical Center, Duarte, CA 91010, USA; kgendzek@coh.org

**Keywords:** KIR, NK cells, haplo-identical HSCT, polymorphism, HLA, leukemia

## Abstract

**Simple Summary:**

We have recently shown a broad disparity of Natural Killer (NK) cell responses against leukemia, highlighting good and bad responders resting on the Killer cell Immunoglobulin-like Receptors (KIR) and HLA genetics. In this study, we deeply investigated KIR2DL NK cell repertoire in combining high-resolution KIR allele typing and multicolor flow cytometry from a cohort of 108 blood donors. Our data suggest that centromeric (cen) AA individuals display more efficient KIR2DL alleles (L1*003 and L3*001) to mount a consistent frequency of KIR2DL+ NK cells and to confer an effective NK cell responsiveness. The transposition of our in vitro observations in T-replete haplo-identical Hematopoietic Stem Cell Transplantation (HSCT) context led us to observe that cenAA HSC grafts limit significantly the incidence of relapse in patients with myeloid diseases after T-replete haplo-identical HSCT. As NK cells are crucial in HSCT reconstitution, one could expect that the consideration of KIR2DL1/2/3 allelic polymorphism could help to refine scores used for HSC donor selection.

**Abstract:**

We have recently shown a broad disparity of Natural Killer (NK) cell responses against leukemia highlighting good and bad responders resting on the Killer cell Immunoglobulin-like Receptors (KIR) and HLA genetics. In this study, we deeply studied KIR2D allele expression, HLA-C recognition and functional effect on NK cells in 108 blood donors in combining high-resolution KIR allele typing and multicolor flow cytometry. The KIR2DL1*003 allotype is associated with centromeric (cen) AA motif and confers the highest NK cell frequency, expression level and strength of KIR/HLA-C interactions compared to the KIR2DL1*002 and KIR2DL1*004 allotypes respectively associated with cenAB and BB motifs. KIR2DL2*001 and *003 allotypes negatively affect the frequency of KIR2DL1^+^ and KIR2DL3^+^ NK cells. Altogether, our data suggest that cenAA individuals display more efficient KIR2DL alleles (L1*003 and L3*001) to mount a consistent frequency of KIR2DL^+^ NK cells and to confer an effective NK cell responsiveness. The transposition of our in vitro observations in the T-replete haplo-identical HSCT context led us to observe that cenAA HSC grafts limit significantly the incidence of relapse in patients with myeloid diseases after T-replete haplo-identical HSCT. As NK cells are crucial in HSCT reconstitution, one could expect that the consideration of KIR2DL1/2/3 allelic polymorphism could help to refine scores used for HSC donor selection.

## 1. Introduction

Natural Killer (NK) cells are granular lymphocytes and form part of the innate immune system. They sense the absence of or decreased HLA class I expression on tumoral or virally infected cells and allogenic cells. They are particularly important in Hematopoietic Stem Cell Transplantation (HSCT), being the first cytotoxic lymphocytes to appear during immune reconstitution before T cell recovery [1], and they are engaged in the beneficial Graft-versus Leukemia (GvL) effect [2,3]. This missing-self recognition by NK cells [4] is mediated through different receptors specific for HLA class I molecules [5]. Among these receptors, Killer cell Immunoglobulin-like Receptors (KIR) play a major role in the functional education of NK cells [6] and the modulation of NK cell functions [7,8]. The absence of inhibitory KIR engagement with its cognate ligand results in triggering KIR^+^ NK cell functions.

The structure and the function of the NK cell repertoire depends on the clonal expression of different KIR combinations defined by the number and the nature of KIR genes and alleles, the HLA class I environment and the immunological history, notably cytomegalovirus (CMV) infection [8,9].

In humans, 15 KIR genes, located on the chromosome 19q13.4, have been identified (Figure 1a) [9]. The number and the nature of KIR genes differ according to individuals allowing the discrimination of two defined KIR haplotypes [10]. The A KIR haplotype is defined by a fixed set of nine KIR genes, including KIR2DS4 as the only activating KIR gene. In contrast, B haplotypes are more diverse, with a variable number of KIR genes, and are characterized by the presence of more than one activating KIR gene and the absence of the KIR2DS4 gene (Figure 1a) [10].

The arrangement of KIR genes and the high sequence similarity facilitate gene gain and loss. Thus, all KIR loci are subject to copy number variation (CNV), particularly in B haplotypes [11]. For instance, the KIR2DL1 gene, present on both A and B haplotypes (Figure 1a), is mainly observed in the form of one copy per haplotype and less than 20% of KIR haplotypes do not display the KIR2DL1 gene [11]. Based on the combination of A and B KIR haplotypes, 660 KIR genotypes were described worldwide (http://www.allelefrequencies.net/). Depending on KIR gene content, centromeric and telomeric KIR motifs were also defined (Figure 1b) [12]. KIR gene length and exon/intron organization vary as illustrated for KIR2DL1/2/3 genes (Figure 1c). Inhibitory KIRs display immuno-receptor tyrosine-based motifs (ITIM) in their cytoplasmic tails (Figure 1d). In contrast, activating KIRs are coupled to adaptor DAP12 protein that contains immunoreceptor tyrosine-based activating motifs (ITAMs). In particular, KIR2DL1 recognizes exclusively HLA-Cw molecules belonging to the group C2 epitope (Lys80) [13] whereas KIR2DL2/3 recognizes HLA-Cw molecules belonging to the C1 group epitope (Asn80) and some HLA-Cw molecules of C2 group (Figure 1d) [14,15].

KIR genes exhibit an allelic diversity, with 1110 KIR alleles currently referenced (https://www.ebi.ac.uk/ipd/kir/). Of note, this allelic polymorphism is more important for the inhibitory rather than for the activating KIR genes. Although some studies have underlined the effect of KIR allelic polymorphism on the phenotype and the function of NK cells as reported for KIR3DL1 [16,17] and for KIR2DL1 [18,19,20], the absence of high-resolution methods limited investigations on KIR^+^ NK cell repertoire. Clinical consequences of KIR allelic polymorphism on KIR diseases were identified for a few KIR genes such as KIR2DL1 for preeclampsia [21], and KIR2DL1 [22], KIR3DL1 [23] for HSCT outcome. Besides KIR allelic polymorphism, HSC donors with a B KIR gene motif are preferentially chosen to decrease relapse incidence after unrelated HLA identical HSCT [12,24], supporting a role of activating KIR or KIR2DL2. This protection against relapse in patients with acute myeloid leukemia (AML) is observed after myeloablative (MAC) and reduced intensity conditioning (RIC) regimens [12,24]. In the context of T-replete HLA haplo-identical HSCT using post-transplant cyclophosphamide (haplo-PTCY), some studies have reported the impact of donor KIR genotypes and/or KIR/HLA mismatches on patient outcome after haplo-PTCY [25,26,27,28,29,30]. In particular, we have shown that KIR2DL/HLA incompatibilities are beneficial to improve patient’s outcome after haplo-PTCY [30]. In haplo-PTCY for which multiple options of HSC donors are feasible, the question arises of whether genetic markers as KIR help to select the best one to improve efficient immune reconstitution commingled with beneficial GvL effect. To address this question, we deeply investigated the influence of KIR2DL1/2/3 allelic polymorphism on the phenotypic and functional structuration of the NK cell repertoire from 108 healthy individuals in combining high-resolution KIR allele typing by Next-Generation Sequencing (NGS) [31], high-resolution flow cytometry using KIR specific mAbs [32] and in silico KIR/HLA-C modelling. In parallel, our in vitro results were compared to ex vivo observations in 81 T-replete haplo-identical HSCT patients treated with PTCY in order to evaluate the predictive dimension of KIR to improve HSC donor selection.

## 2. Results

### 2.1. Predominant KIR2DL1*003 and KIR2DL3*001 Alleles Are Frequently Identified as Unique Allele and Are Associated with Centromeric AA Motifs

In this study, we focused our investigations on KIR2DL1/2/3 allelic diversity as their gene frequencies are close to 100%, as well as because they are the most engaged inhibitory receptors with HLA-C ligands. On 97 KIR2DL1^+^ genotyped healthy individuals, 5 KIR2DL1 alleles were identified. The most frequent KIR2DL1 allele was KIR2DL1*003 (39.7%), followed by *002 (33.3%), *004 (20.7%), *001 (2.5%) and *007 (2.5%). Interestingly, KIR2DL1*001 and *004 were more frequently associated with another KIR2DL1 allele than found alone (Figure 2a). In contrast, KIR2DL1*003 was predominantly found as unique (Figure 2a).

Besides KIR2DL1, the allelic polymorphism of both KIR2DL2 and 2DL3, being alleles of the same gene, was investigated from the same individuals. On 92 KIR2DL2/3^+^ genotyped individuals, 11 KIR2DL2/3 alleles were identified, L3*001 (30.4%), L3*002 (26.1%), L2*001 (18.5%) and L2*003 (11.4%) being the most frequent ones. KIR2DL2*001 and L3*002 were overrepresented combined with another KIR2DL2/3 allele rather than alone (Figure 2b). In contrast, KIR2DL3*001, and to a lesser extent L3*003, were predominant as unique KIR2DL3 allele (Figure 2b). In contrast to KIR2DL1/2/3 genes, activating KIR2DS1 and KIR2DS2 genes whose frequencies were 37.0% and 55.6%, respectively, showed only one predominant allele (S1*002, 93.6% and S2*001, 97%).

Since KIR2DL1/2/3/S1/S2 genes belong to different KIR haplotypes with distinctive centromeric (cen) and telomeric (tel) motifs, we further evaluated KIR2D allele frequencies in regards to cen and tel motifs. KIR2DL1*003 was predominant in cenAA individuals with increasing incidence when associated with telB+ (AB/BB) compared to telAA motifs (Figure 2c). In contrast, KIR2DL1*002 was predominant in cenAB individuals, with decreasing frequency with telB+ compared to telAA motifs (Figure 2c). KIR2DL1*004 or the absence of KIR2DL1 characterized cenBB individuals (Figure 2c). KIR2DL3*001 was predominant in cenAA individuals (Figure 2d). KIR2DL3*001, L3*002, L2*001 and L2*003 were similarly represented in cenAB individuals, with increasing incidence of L2*001 and decreased incidence of L2*003 in telB+ compared to telAA motifs (Figure 2d). The predominance of KIR2DL2*003 characterizes cenBB donors (Figure 2d).

KIR2DL1/2/3/S1/S2 allele typing led us to determine 59 allele combinations, some of them being found in several individuals (*n* = 18) and others unique (*n* = 41) (Appendix A). The most frequent KIR2D allele combination found in 9 cenAA individuals displayed a linkage disequilibrium (LD) between L1*003 and L3*001, preferentially in a C1^+^ environment (Appendix A). The second one found in 6 cenAB individuals also showed this LD and more specifically the L1*004-L2*001 combination, in a preferential C2^+^ environment (Appendix A). The L1*002-L3*002 combination was observed in both cenAA and AB individuals with different KIR2D associations (Appendix A).

### 2.2. Predominant KIR2DL2 Alleles Associated with Centromeric B+ Motifs Negatively Affect Both the Frequency and the Expression Level of KIR2DL1 Allotypes

Due to the limited KIR2DS allelic polymorphism, we furthered our investigations on inhibitory KIR2DL1/2/3 expression. KIR2DL1^+^ NK cell frequency was determined using the combination of KIR2DL1 and 2DL1/S1 specific mAbs, as illustrated in Figure 3a for one representative KIR2DL1/S1^+^ individual. KIR2DL1^+^ NK cell frequency was lower in cenAB/BB (cenB+) compared to cenAA individuals (Figure 3b). As the KIR2DL2 gene may supersede KIR2DL1^+^ NK cell functions [33], we evaluated its influence. Both L2*001 and L2*003 alleles negatively affected KIR2DL1^+^ NK cell frequency (Figure 2b). The frequency of KIR2DL1*004^+^ NK cells was lower compared to L1*002^+^ NK cells or L1*003^+^ NK cells (Figure 3c). A decrease of both KIR2DL1*002^+^ and L1*003^+^ NK cell frequencies was observed in cenB^+^ vs. cenAA individuals due to the presence of KIR2DL2 (Figure 3c). In particular, the frequency of KIR2DL1*002^+^ NK cells was lower in L2*003+ compared to L2- individuals (Figure 3c).

To determine the KIR2DL1 expression level for main KIR2DL1 allotypes, we used a combination of 8C11 and 1A6 mAbs [32]. The 8C11 recognizes KIR2DL1 allotypes with P154 amino acid (L1*002 and L1*003) and 1A6 recognizes KIR2DL1 allotypes with P114 amino acid (L1*002 and L1*004) (Figure 3d). Thus, in gating on KIR2DL1^+^/2/3/S1/2^−^ NK cells, we were able to further discriminate NK cell subsets expressing L1*002 (8C11^+^ 1A6^+^), L1*003 (8C11^+^ 1A6^−^) or L1*004 (8C11^−^ 1A6^+^) allotype (Figure 3e). Of note, we did not assess the expression of the KIR2DL1*002 allotype for L1*002, *003^+^ and L1*002, *004^+^ individuals as potential co-expression of both allotypes can be observed. From 11 KIR2DL1*003, *004 donors, we confirmed that the co-expression of both KIR2DL1 allotypes is limited and that KIR2DL1*003^+^ NK cells are the most representative subset (Figure 3f). The MFI of KIR2DL1*004 was lower compared to those of L1*002 and L1*003 allotypes (Figure 3g,h). Interestingly, KIR2DL2*001 negatively affected on the level expression of the KIR2DL1*003 allotype (Figure 3h).

### 2.3. The KIR2DL2 Gene Specific of Centromeric B+ Motifs Positively Affects the Expression Level of the Dominant KIR2DL3*001 Allotype

To investigate the expression of KIR2DL3 allotypes, we used a combination of mAbs leading to firstly discriminate 2DL2 and 2DL3 and secondly to identify the KIR2DL3*005 allotype (Figure 4a). Thus, in gating on KIR2DL2/3/S2^+^ NK cells, different subsets are identified following KIR gene content (Figure 4b). In KIR2DL3^+^/2^−^/S2^−^ individuals, only KIR2DL3^+^ NK cells are observed. In KIR2DL2/3/S2^+^ individuals, 3 populations are observed: L3^+^/L2^+/−^/S2^+/−^, L3^−^/L2^+^/S2^−^ and L3^−^/L2^+/−^/S2^+^. Finally, as the KIR2DL3*005 allotype was not recognized by the anti-KIR2DL3 specific mAb, it is possible to discriminate this allotype (Figure 4b). CenB^+^ motifs linked to the presence of L2*001 or L2*003 alleles constrained KIR2DL3^+^ NK cell frequency (Figure 4c). The frequency of KIR2DL3*001^+^ NK cells was higher compared to KIR2DL3*002^+^ NK cells (Figure 4d), but their expression levels were similar (Figure 4e). However, the expression level of the KIR2DL3*001 allotype was higher in the presence of KIR2DL2 (Figure 4e). The expression level of KIR2DL3*005 was higher compared to L3*001, *002 and *007 allotypes (Figure 4f).

### 2.4. CenAA KIR2DL Allotypes Are Associated with an Efficient NK Cell Responsiveness

The degranulation of KIR2DL1^+^ NK cells was assessed ex vivo against the standard HLA class I negative 721.221 B-EBV cell line (221). For that purpose, the KIR2DL1^+^/S1^−^ NK (CD3^−^ CD56^+^) cell subpopulation was first gated then KIR2DL2/3/S2 expression was excluded (Figure 5a). Using the combination of both KIR2DL1 specific 8C11 and 1A6 mAbs [32], NK cell subsets expressing 2DL1*002 (8C11^+^ 1A6^+^), L1*003 (8C11^+^ 1A6^−^) and L1*004 (8C11^−^ 1A6^+^) allotypes were discriminated (Figure 5a). As expected for licensed NK cells, the degranulation of KIR2DL1^+^ NK cells was higher for C2^+^ than for C2^-^ individuals (Figure 5b). Both KIR2DL1*002^+^ NK cells and L1*003^+^ NK cells have a stronger degranulation compared to L1*004^+^ NK cells in C2^+^ donors (Figure 5c).

In parallel, we evaluated the ex vivo degranulation of KIR2DL2^+^ and L3^+^ NK cells assessed against the 221-cell line. KIR2DL2/3/S2^+^/L1/S1^−^ NK cell subset was targeted using the previous combination of anti-KIR2DL3/S2 [32] and anti-KIR2DL3 mAbs to isolate the KIR2DL2^+^ NK cell subset from individuals harboring a unique KIR2DL2 allotype or KIR2DL2/3 allele combination (Figure 5d). We did not observe a difference of educated KIR2DL2/3^+^ NK cell degranulation between cenAA and AB individuals being respectively L3^+^/L2^−^ and L3^+^/L2^+^ (Figure 5e). However, while KIR2DL2/3^+^ NK cell degranulation was homogeneous in cenAA individuals, the quartiles of KIR2DL2/3^+^ NK cells were very outlying in cenAB individuals suggesting that other parameters may modulate this response. KIR2DL2*001^+^ NK cells observed mainly in cenAB individuals showed the lowest degranulation potential compared to L3*001^+^, L3*002^+^ and L3*005^+^ NK cells in C1^+^ individuals (Figure 5f). No difference can be highlighted between KIR2DL3 allotypes due to the limited size number of L3*005 and L3*002 individuals. Of note, KIR2DL2*003^+^ NK cell degranulation frequencies associated with cenBB motifs were higher even though disparate.

### 2.5. Broad Disparity of HLA-C Recognition between Studied KIR2DL Allotypes

KIR and HLA class I allelic polymorphism may directly affect molecular interactions between KIR and their corresponding ligands modulating the inhibition of NK cell degranulation. To test this hypothesis, we further investigated whether KIR2DL1/2/3 and/or HLA-C allele specificities modulate the inhibition of KIR2DL^+^ NK cell degranulation. Peripheral blood mononuclear cells were stimulated to amplify the NK cell compartment in vitro (Figure 6a). After 2 weeks culture, NK cell degranulation was evaluated against a panel of HLA-C transfected 221 cell lines. KIR2DL1^+/^2^−^/3^−^/S1^−^/S2^−^/NKG2A^−^ (L1^+^ others^−^) were targeted and the 1A6/8C11 mAbs combination led us to discriminate different NK cell subsets expressing L1*002, *003 or *004 allotype in C2^+^ individuals (Figure 6b). In parallel, KIR2DL2/3^+^/1/S1^−^/S2^−^/NKG2A^−^ (L3^+^ others^-^) NK cell subsets were targeted and the KIR2DL3/S2 (1F12)/2DL3 mAbs combination led us to discriminate L3^+^ and L2^+^ NK cell subsets (Figure 6c).

We assessed the inhibition of KIR2DL^+^ NK cell degranulation against a panel of 3 C1 and four C2 transfected 221 cell lines focusing on main KIR2DL1/2/3 allotypes. The C2 specificity of KIR2DL1^+^ NK cells was confirmed with nonetheless a hierarchy of recognition toward different HLA-C molecules belonging to the C2 group (data not shown [34]). In contrast, 2DL2 and 2DL3 did not exclusively recognize C1 ligands since a strong inhibition of 2DL2^+^ and 2DL3^+^ NK was observed with the 221-HLA-C*04:01 cell line, being C2^+^ target (data not shown [34]) as we previously reported [15].

The highest inhibition level of NK cell degranulation was observed for the L1*003 allotype for all C2^+^ targets and more specifically against the 221-C*02:02 between L1*002^+^ and L1*003^+^ NK cells and the 221-C*15:03 target between L1*003^+^ and L1*004^+^ NK cells (Figure 6d). Using an in-silico modelling of KIR2DL1 allotypes (Appendix B), we observed that the 5 polymorphic residues in the extracellular D1 and D2 domains (Appendix A) does not generate any major conformational modifications (Appendix A). Thus, these polymorphic positions cannot explain that the KIR2DL1*003 allotype exhibits a trend of a higher inhibition level of NK cell degranulation against the 221-HLA-C*04:01 cell line compared to its counterparts (Figure 6d). Considering KIR2DL2/3 allelic polymorphism, heterogeneous levels of inhibition of KIR2DL2/3^+^ NK cell degranulation depending on KIR2DL2/3 allotypes was observed mainly against less stringent ligand as HLA-C*08:02 (C1) and -C*02:02, -C*06:02 (C2) showing a better inhibition even not significant of KIR2DL2 than KIR2DL3 allotypes (Figure 6d).

### 2.6. Beneficial Impact of cenAA HSC Donors on Relapse Incidence after T-Replete Haplo-Identical HSCT in Myeloid Diseases

Our previous observations support that cenAA L1*003 and L3*001 allotypes are associated with a higher frequency of KIR2DL^+^ NK cells and better responsiveness. We further determine the centromeric KIR gene motifs of HSC donors. On 81 included haplo-PTCY patients, 39 cenAA and 42 cenAB.BB (cenB^+^) donors were identified (Appendix A). Haplo-PTCY performed from cenAA or cenB^+^ donors were identical for the clinical characteristics including diseases, status at treatment, disease risk index and conditioning (Appendix A). The relapse incidence between haplo-PTCY performed from cenAA vs. cenB^+^ donors did not differ when all patients were included (Figure 7a). However, relapse incidence was decreased with cenAA donors were compared to cenB^+^ HSC donors for patients with myeloid diseases (Figure 7b) but not for patients with lymphoid diseases (Figure 7c).

To assert that cenAA HSC grafts significantly limit relapse incidence in patients with myeloid malignancies after haplo-PTCY, we performed univariate and multivariate analyses including as variables age, recipient gender, status at treatment, disease risk index (DRI), conditioning and donor HSC cenAA motif. Univariate analysis identified DRI (high/very high vs. intermediate) as the most significant factor predicting relapse (HR = 4.07 [95%CI 1.87–8.89], *p* = 0.0004). There was a trend for a lesser relapse rate in patients with myeloid diseases grafted with cenAA HSC donors (HR = 0.50 [95%CI 0.23–1.09], *p* = 0.08). Age, gender, status at treatment and conditioning were not significantly associated with relapse (data not shown [34]). Multivariate analysis confirmed that donor HSC cenAA motifs were significantly associated with decreased relapse in contrast to DRI that is associated with increased relapse in patients with myeloid diseases after haplo-PTCY (Table 1). Overall, these results sustain a beneficial effect of cenAA donors on relapse incidence after haplo-PTCY only in myeloid diseases, arguing for a better GvL effect driven by NK cells.

## 3. Discussion

We confirmed a limited number of alleles for each KIR2DL receptor with closed frequencies reported in other European cohorts [35] and an association with KIR A (L*001, *002 and *003 alleles) and B (L1*004) haplotypes [21,36,37,38]. For the first time, we highlighted the dominance of L1*003 and L3*001 alleles in cenAA individuals. Moreover, we showed that L1*002, L3*002 and L2*001 alleles were predominant in cenAB whereas L1*004 or no 2DL1 and L2*003 alleles were predominant in cenBB individuals. We refined some LD [39,40], taking into account 2DL1/2/3/S1/S2 allele combinations encountered in both cen and tel motifs, although haplotype family segregation was lacking in our study. Thus, the clustering taking into account only cenA and B motifs is probably not completely accurate.

KIR2DL1/2/3 allotypes confer different phenotypic and functional characteristics to NK cells based on KIR cen motifs. CenAA individuals display mainly the L1*003 allotype with highest frequency, expression level [19,20,21,38] and strength KIR/HLA-C interactions compared to the L1*002 allotype in cenAB and to the L1*004 allotype in cenBB individuals. In this study, we did not determine the CNV for KIR2DL1. However, although the frequency of NK cells expressing a given KIR correlates with the CNV of that gene [38], the coexpression of multiple copies is infrequent [41]. Thanks to the combination of KIR specific 1A6 and 8C11 mAbs [32], we discriminate for the first time the main KIR2DL1 allotypes. We demonstrated that KIR2DL1^+^ NK cell degranulation was higher with the L1*003 allotype, usually associated with cenAA motifs.

The stronger recognition of C2 ligands by KIR2DL1^+^ NK cells was observed for the KIR2DL1*003 allotype as previously observed [18]. In contrast to other studies pooling KIR2DL1*002 and *003 allotypes with R245 residue [18,20], we were able to discriminate both KIR2DL1 allotypes showing that the L1*002 allotype (R245) interacts with all the C2^+^ targets similar with the L1*004 allotype (C245). The modelling of the studied KIR2DL1 allotypes associated with the HLA-Cw4 molecule [13] suggests that the difference in functionality between the allotypes is not due to the polymorphic positions located on the domains binding of the HLA-C molecule. However, we cannot exclude an influence of loaded peptides, which are probably different in our in vitro cellular model and in in silico modelling. Although a minimal role of peptide has been deduced from the crystal structure of KIR2DL1/HLA-Cw4 complex [13], studies with synthetic peptide analogs have showed that substitution of Lys8 in the peptide with acid residue results in KIR binding loss [42,43].

The KIR2DL1*004 allele associated with cenBB motifs confers a lower degranulation potential to NK cells. It negatively affects the frequency of KIR2DL1^+^ and 2DL3^+^ NK cells and the level expression of L1*003 and *002 allotypes. In contrast, it positively affects the level expression of L3*001 allotype when associated with cenAB motifs. In absence of KIR2DL2 specific mAb, it is difficult to grasp KIR2DL2^+^ NK cell frequency and the expression level of KIR2DL2. We observed that KIR2DL2^+^ NK cell degranulation was more heterogeneous than L3^+^ NK cells without identifying link with specific genetic parameters. Educated KIR2DL2*001^+^ NK cells harbored the worst degranulation compared to all KIR2DL2/3 allotypes. The L2*001 allele is often observed in cenAB individuals in combination with L3*001 or L3*002 alleles whereas L2*003 is mainly observed in cenBB donors without KIR2DL3. This observation suggests that KIR2DL3 could negatively regulate the function of the L2*001 allotype in cenAB individuals.

Heterogeneous levels of inhibition of KIR2DL2/3^+^ NK cell degranulation was observed mainly against less stringent ligands as HLA-C*08:02 and -C*02:02, -C*06:02 showing a better inhibition even not significant of 2DL2 than L3 allotypes. HLA-C allelic polymorphism could affect the KIR2DL2/3/HLA-C affinity and the functionality of NK cells, both from a conformational and on the level of expression of this ligand [44,45]. The peptide presented by HLA-C molecules could also affect the KIR2DL2/3-ligand affinity [46]. In contrast to KIR2DL1, the peptide modulates the binding of KIR2DL2/3 to the HLA-Cw3 through direct interactions [14,42]. Studies have also shown that KIR2DL2/3 have a higher affinity with the HLA-C*03:04 allotype loaded with Hepatitis Chronic Virus (HCV)-derived peptides [46]. Altogether, we suggest that cenAA individuals display more efficient KIR2DL alleles (L1*003 and L3*001) to mount a consistent frequency of KIR2DL^+^ NK cells and to confer an effective spontaneous degranulation of NK cells. Nonetheless, further investigations on a broader cohort would be necessary to include rare KIR2DL alleles not documented in our study. Moreover, our functional results raise additional questions concerning the influence of HLA-C polymorphism. Indeed, we observed a broad spectrum of HLA-C recognition by KIR2DL independently of C1/C2 classification, suggesting an impact of HLA-C polymorphism on the phenotypic and functional structuration of the NK cell repertoire. Finally, we did not include peptide influence that seems to modulate KIR2DL affinity.

Some studies have reported either a beneficial [25,28] or no impact [26] of donor KIR B genotypes on relapse incidence after haplo-PTCY. However, the impact of donor KIR centromeric motifs on clinical outcome was not investigated so far after T-replete haplo-PTCY. Here, we report with multivariate analysis that myeloid patients grafted with HSC donors harboring a KIR cenAA motif have a lower incidence of relapse compared to cenB^+^ donors after haplo-PTCY. In addition, multivariate analysis showed that DRI was the most significant factor affecting relapse incidence in myeloid patients after haplo-PTCY, as previously reported [47,48]. By contrast, no protective effect of donor B^+^ genotype was shown on relapse incidence (data not shown [34]). Heterogeneities concerning the proportion of AML patients, conditioning regimen and stem cell source between published studies [25,26,28] and that reported here could explain these discordances. The protective effect of cenAA donors we report here was observed whatever the immunosuppressive regimen and only in myeloid patients, including 59% of AML patients. This is in agreement with our recent work showing that KIR^+^ NK cell subsets are preferentially engaged against AML [49]. Overall, these clinical data suggest that the GvL effect could be driven by KIR2DL1 or 2DL3 specific of cenAA motifs after haplo-PTCY.

## 4. Materials and Methods

### 4.1. Healthy Individuals

One hundred and eight blood donors were recruited at the Blood Transfusion Center (Etablissement Français du Sang, Nantes, France) and informed consent was given by all donors. Preparation and conservation of these biocollections have been declared to French Research’s Minister (DC-2014-2340) and has received approval from the IRB (2015-DC-1).

### 4.2. Cohort of T-Replete Haplo-Identical HSCT Patients

This retrospective study has included 81 adult patients with various hematological malignancies who received a T cell-replete haploidentical HSCT with post-transplant cyclophosphamide (haplo-PTCY) in the Hematology Department of Nantes University Hospital. Conditioning regimens consisted of a Baltimore-based RIC regimen with fludarabine (*n* = 26) [50] or clofarabine (Clo-Baltimore, *n* = 27) [51] or a CloB2A1 regimen with clofarabine 30 mg/m^2^/day (d), d-6 to -2, busulfan 3.4 mg/kg d-3 and d-2, and ATG 2.5 mg/kg d-1 (*n* = 28) [52]. The source of graft was peripheral blood stem cell (PBSC) for all cases. The study complies with the Declaration of Helsinki. All patients provided informed consent for collecting their own data from the PROMISE database of the European Bone Marrow Transplantation (EBMT). This study was approved by the Ethics Review Board of the Nantes University Hospital and all patients and HSC donors provided informed consent. The clinical outcome and immune reconstitution of some patients have been previously reported [30,53] and have been updated in October 2020 for this study. The primary endpoint was relapse incidence.

### 4.3. Cells (PBMCs and Cell Lines)

Peripheral Blood Mononuclear Cells (PBMC) were isolated from healthy individuals using Ficoll-Hypaque (Biosera, Nuaille, France). NK cell amplification was done using an in vitro model based previously described [54]. HLA class I-deficient 721.221 lymphoblastoid cells, referred to as 221 cells, were used as positive control to assess NK cell degranulation. HLA-C*03:04 (C1), -C*07:01 (C1), -C*08:02 (C1), -C*02:02 (C2), -C*04:01 (C2), -C*06:02 (C2) and -C*15:03 (C2) transfected 221 cells were used to evaluate KIR2DL1/2/3^+^ NK cell degranulation as previously described [15]. The 221-cell line and HLA-C-transfected 221 cell lines were cultured in RPMI 1640 medium (Life Technologies, Carlsbad, CA, USA) containing glutamax (Life Technologies) and penicillin-streptomycin (Life Technologies), and supplemented with 10% FBS (Life Technologies). Mycoplasma tests performed by PCR were negative for all cell lines.

### 4.4. HLA Class I and KIR Genotyping

HLA-A, -B, and -C typing was carried out by Next-Generation-Sequencing (NGS) using Omixon Holotype HLA^®^ (Omixon, Budapest, Hungary). Generic KIR typing was performed on all individuals and HSC donors using a KIR multiplex PCR-SSP method [55]. Centromeric and telomeric KIR motifs were defined taking into account KIR2DL2/3/S2 and KIR3DL1/S1/2DS1/2DS4 genes respectively as reported [12]. KIR2DL1/2/3/S1/S2 alleles were assigned on all healthy individuals by NGS [31]. KIR genes were firstly captured by Long Range PCR using five intergenic KIR primers according the LR-PCR protocol already described [31]. Qubit dsDNA high sensitivity Assay kit (Life technologies, Villebon sur Yvette, France) was used to quantify the starting DNA library on a Qubit^®^ fluorometer (Life technologies). The KIR library preparation was performed using the NGSgo GENDX kit (Bedia Genomics, Chavenay, France) according to the manufacturer’s instructions. The final denatured library was subsequently sequenced by using a MiSeq sequencer (HLA laboratory, EFS Nantes, France) with 500 cycles v2 kits which generated 250-base paired-end sequence reads. The quality of raw data sequences was monitored by using the Sequencing Analysis Viewer (SAV) Illumina software. KIR2DL1/2/3/S1/S2 allele assignment was performed by using the Profiler software developed by Dr M. Alizadeh (Research Laboratory, Blood Bank, Rennes, France) [31]. An updated KIR allele library (v2.9) was implemented into the Profiler software version 2.24.

### 4.5. Phenotypic Analysis and CD107a Mobilization Assay Using Flow Cytometry

The phenotype of NK cells, defined as CD3^-^CD56^+^, was determined by 8-color multiparameter flow cytometry (MFC) using the following mAbs: anti-KIR2DL1-PE (143211; R&D Systems, Minneapolis, MN, USA), anti-KIR2DL1/S1-PE/-PC7 (EB6), anti-KIR2DL2/3/S2-PC7/-PerCP5.5 (GL183; Beckman Coulter Immunotech, Marseille, France), anti-KIR2DL3-FITC (clone 180701; R&D systems), anti-CD3-APC-Cy7 (SK7, Sony Biotechnology, San Jose, CA, USA) and CD3-PerCP (SK7, BD Biosciences, Le pont de Claix, France), anti-NKG2A-PE/-PerCP (Z199; Beckman Coulter), anti-CD56-APC-Cy7/-BV510 (HCD56), anti-KIR2DL1/2/3/S2-AF647 (8C11), anti-KIR2DL3/S2-AF647 (1F12), and anti-KIR2DL1/2/3/S1/S2-FITC (1A6) generated and characterized in our laboratory [32]. Ex vivo NK cells or in vitro expanded NK cell degranulation was assessed as previously reported [53]. MFC data were collected on a FACS Canto II instrument (BD Biosciences) and analyzed with Flowjo^TM^ 10.2 software (LLC, Ashland, OR, USA).

### 4.6. Statistical Analysis

Categorical data were analyzed by Chi-square test and univariate comparisons were performed by the Student t test. Statistical differences in KIR2DL^+^ NK cell frequencies between individuals having different KIR2DL1/2/3 alleles were analyzed with unpaired t tests or one-way ANOVA test for multiple comparisons using the GraphPad Prism v6.0 software (San Diego, CA, USA). Clinical and demographic variables for patients (i.e., age, gender, status at treatment, disease risk index, conditioning and donor centromeric AA KIR motif) were evaluated for their impact on relapse incidence in univariate and multivariate analyses using LogRank test and Cox proportional hazards models adjusted for significant clinical factors. Multivariate analysis was performed including only variables having a p-value less than 0.20. Univariate and multivariate analyses were performed with the Medcalc (Ostend, Belgium) software. *p* values < 0.05 were considered statistically significant.

## 5. Conclusions

Altogether, our data suggest that cenAA individuals display more efficient KIR2DL alleles (L1*003 and L3*001) to mount a consistent frequency of KIR2DL^+^ NK cells and to confer an effective NK cell responsiveness. The transposition of our in vitro observations in T-replete haplo-identical HSCT context led us to observe that cenAA HSC grafts limit significantly the incidence of relapse in patients with myeloid diseases after T-replete haplo-identical HSCT. Nevertheless, our conclusions have to be taken with caution and to be confirmed from a larger cohort of haplo-identical HSCT donor/recipient pairs. NK cell characteristics are crucial in HSCT, one could expect that the consideration of KIR2DL1/2/3 allelic polymorphism could help to refine scores used for HSC donor selection, and to evaluate its influence on HSCT outcome.

## Figures and Tables

**Figure 1 cancers-12-03595-f001:**
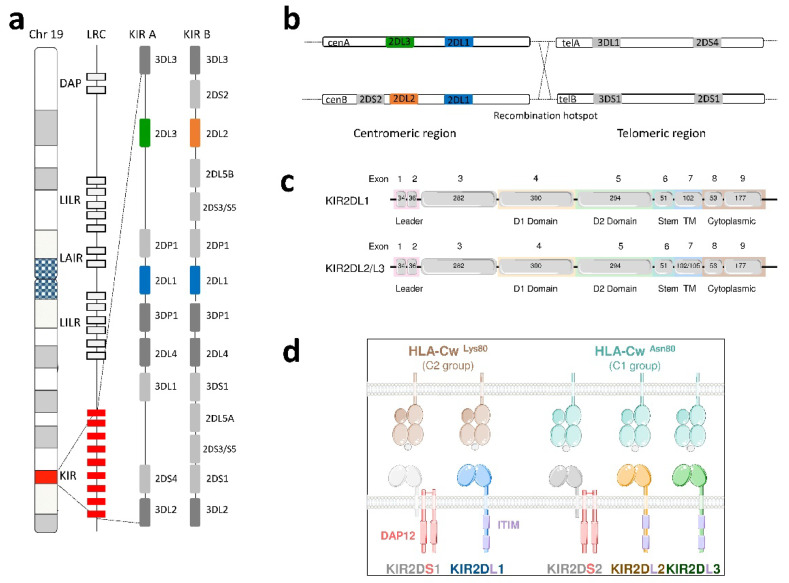
Localization, organization of Killer cell Immunoglobulin-like Receptors (KIR) genes and HLA-C specificity of KIR2DL interactions. KIR genes are located on chromosome 19 in the Leucocyte Region Complex (LRC), which also contains genes encoding DAP adaptor protein and other Natural Killer (NK) cell receptors. The A KIR haplotype is defined by a fixed set of nine genes with only 2DS4 as activating genes, whereas B haplotypes are more diverse and characterized by the presence of more than one activating KIR gene and the absence of 2DS4 (**a**). Different KIR centromeric (cen) and telomeric (tel) KIR motifs are defined depending on the presence or absence of KIR2DL3/L2/S2 and KIR2DS1/S4/3DL1/S1 genes respectively (**b**). KIR2DL1/L2/L3 genes have eight exons, as well as a pseudoexon 3 coding for two Ig-like domains and a long intracytoplasmic tail (**c**). KIR2DL1/L2/L3/S1/S2 receptors interact with specific HLA-C molecules divided into C1 or C2 group (**d**).

**Figure 2 cancers-12-03595-f002:**
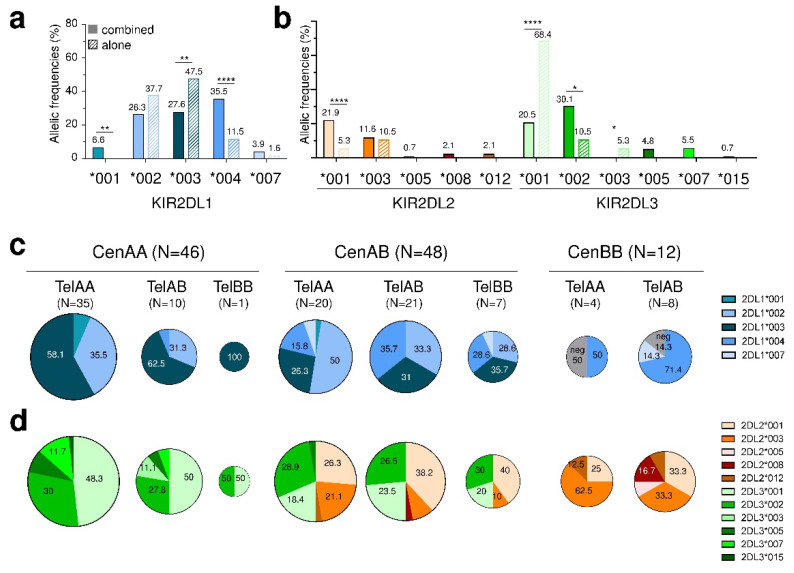
Predominant KIR2DL1*003 and KIR2DL3*003 alleles are frequently identified as unique allele and are associated with specific centromeric motifs. (**a**) KIR2DL1 allelic frequencies (%) established in blood donors either found alone (clear bars, *n* = 59) or combined with another KIR2DL1 allele (filled bars, *n* = 38). (**b**) KIR2DL2/3 allelic frequencies (%) established in blood donors either found alone (clear bars, *n* = 19) or combined with another KIR2DL2/3 allele (filled bars, *n* = 73). (**c**) KIR2DL1 and (**d**) KIR2DL2/3 allele frequencies (%) established in blood donors classified according to their centromeric (cenAA, AB, BB) and telomeric (telAA, AB, BB) motifs. The size of each pie chart varies according to sample size. Only KIR2DL1/2/3 allele frequencies higher than 10% are indicated on pie charts. KIR2DL1/2/3 alleles were assigned by Next-Generation-Sequencing technology using Profiler software as described in the Materials and Methods section. Specific colors were used to discriminate each KIR2DL1/2/3 allele. Statistical differences were analyzed using chi^2^. * *p* < 0.05, ** *p* < 0.01, **** *p* < 0.0001.

**Figure 3 cancers-12-03595-f003:**
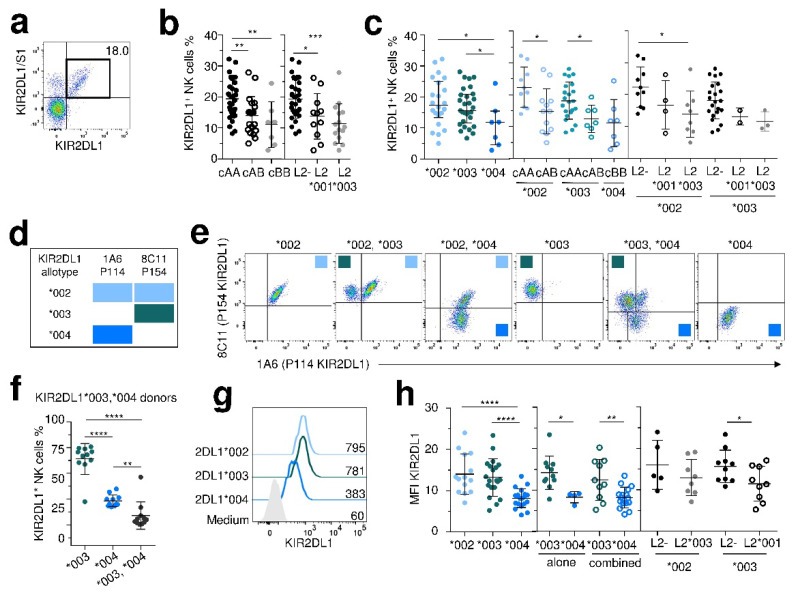
Predominant KIR2DL2 alleles associated with centromeric B+ motifs negatively affect both the frequency and the expression level of KIR2DL1 allotypes. (**a**) Density plot illustrating the cell targeting strategy used to target KIR2DL1 allotypes on NK (CD3^−^, CD56^+^) cells by combining anti-KIR2DL1/S1 (EB6) and a specific anti-KIR2DL1 mAbs. (**b**) Scatter plots showing the frequencies of KIRDL1^+^ NK cells in individuals with centromeric (c) cAA (*n* = 32), cAB (*n* = 19), cBB (*n* = 7) motifs, and in 2DL2^−^ (*n* = 32), L2*001+ (*n* = 11) and L2*003+ (*n* = 15). (**c**) Scatter plots showing the frequencies of KIRDL1^+^ NK cells in individuals with only one 2DL1 allele (L1*002^+^, *n = 22*, L1*003^+^, *n = 29* and L1*004^+^, *n* = 7), in cAA (L1*002, *n* = 10, L1*003, *n* = 22), cAB^+^ (L1*002, *n = 12*, L1*003, *n* = 7) and cBB (L1*004, *n* = 7), in L1*002+/L2- (*n* = 10), L1*002/L2*001+ (*n = 4*), L1*002/L2*003+ (*n* = 8), L1*003+/L2^−^ (*n* = 22), L1*003/L2*001+ (*n* = 2) and L1*003/L2*003+ (*n* = 3) individuals. (**d**) Site recognition of specific anti-KIR2D 1A6 and 8C11 mAbs and corresponding binding to main KIR2DL1 allotypes. (**e**) Density plots illustrating the overall strategy used to target KIR2DL1 allotypes on NK cells in 6 representative individuals using 1A6 and 8C11 mAbs combination. (**e**) Squares with the blue-scale gradient used in (**d**) lead to identify NK cell subset expressing each KIR2DL1 allotype. (**f**) Scatter plots showing the frequency of KIRDL1^+^ NK cells in 11 L1*003, *004 individuals. (**g**) MFI of KIR2DL1 on NK cells depending on KIR2DL1 allotype. (**h**) Relative MFI of KIR2DL1 represents the ratio of the MFI of KIR2DL1 on the MFI of the isotype control for each KIR2DL1 allotype (L1*002^+^, *n* = 14, L1*003^+^, *n* = 20 and L1*004^+^, *n* = 20), as unique KIR2DL1 allele (alone, L1*003, *n* = 11, L1*004, *n* = 4) or combined with another KIR2DL1 allele (combined, L1*003, *n = 10*, L1*004, *n* = 16), and in L2^-^ (L1*002^+^, *n* = 5, L1*003, *n* = 10), L2*003+ (*n* = 8) or L2*001+ (*n* = 9) individuals. * *p* < 0.05, ** *p* < 0.01, *** *p* < 0.001, **** *p* < 0.0001.

**Figure 4 cancers-12-03595-f004:**
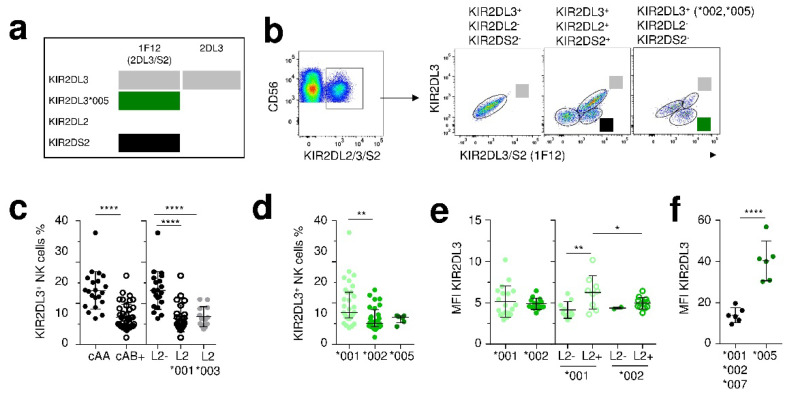
The KIR2DL2 gene specific of centromeric B+ motifs positively affects the expression level of the dominant KIR2DL3*001 allotype. (**a**) KIR2DL2/3/S2 recognition of anti-KIR2DL3 mAbs and an anti-KIR2DL3/S2 (1F12) mAbs. (**b**) Density plots illustrating the overall strategy used to target KIR2DL3 allotypes on NK (CD3^-^ CD56^+^) cells in three representative individuals (L3^+^/L2^+^/S2^+^, L3^+^/L2^−^/S2^−^ and heterozygous L3*005, L3*002/L2^−^/S2^−^ by combining anti-KIR2DL2/3/S2 (GL183), anti-KIR2DL3 and anti-KIR2DL3/S2 (1F12) mAbs. KIR2DL3^+^ NK cells are indicated by a light grey square, KIR2DS2^+^ NK cells are indicated by a black square and KIR2DL3*005^+^ NK cells are indicated by a dark green square. **(c)** Scatter plots showing the frequencies of KIR2DL3^+^ NK cells in cenAA (filled circles, *n* = 21) vs. cenAB (*n* = 41) individuals and in KIR2DL2^−^ (filled black circles, *n* = 21) vs. L2*001+ (clear circles, *n* = 26) or L2*003+ (filled grey circle, *n* = 13) individuals. (**d**) Frequencies of KIR2DL3^+^ NK cells following KIR2DL3 allotypes (L3*001, *n* = 31, L3*002, *n* = 28, and L3*005, *n* = 6). (**e**) Relative MFI of KIR2DL3 represents the ratio of the MFI of KIR2DL3 on the MFI of the isotype control for each KIR2DL3 allotype (L3*001^+^, *n* = 19, and L3*002^+^, *n* = 14), in either L2^−^ (filled circle, L3*001, *n* = 10, L3*002, *n* = 2), or L2^+^ (clear circle, L3*001, *n* = 9, L3*002, *n* = 12) individuals. (**f**) Relative MFI of KIR2DL3^+^ NK cells for 6 individuals expressing KIR2DL3*005 allotype in combination with L3*001, *002 or *007 allotypes. The MFI was assessed for KIR2DL3*005 allotype or other KIR2DL3 allotype (L3*001, *002 or *007) with anti-KIR2DL3/S2 (1F12) mAb. Statistical differences between different groups were analyzed using unpaired t-tests or one-way ANOVA followed by turkey’s multiple comparison test. * *p* < 0.05, ** *p* < 0.01, **** *p* < 0.0001.

**Figure 5 cancers-12-03595-f005:**
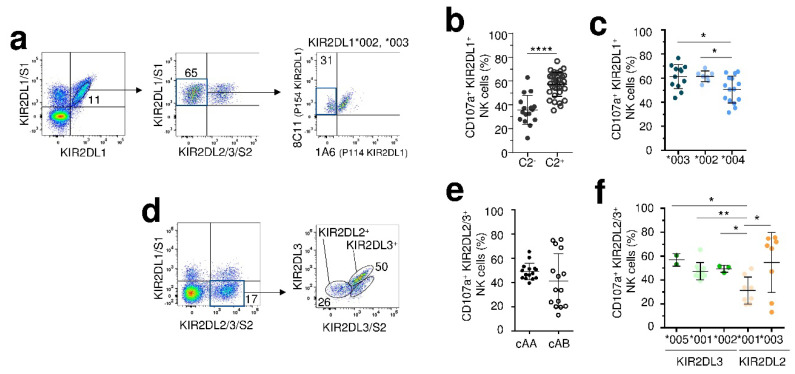
CenAA KIR2DL allotypes are associated with an efficient NK cell responsiveness. (**a**) Density plots illustrating the targeting strategy of KIR2DL1^+^/2/3/S1/S2^−^ NK cells. A first combination of anti-KIR2DL1 and anti-KIR2DL1/S1 (EB6) leads to target KIR2DL1^+^ KIR2DS1^-^ NK cells. A second combination leads to eliminate KIR2DL2/3/S2^+^ NK cells using the anti-KIR2DL2/3/S2 (GL183) mAb. Finally, each KIR2DL1 allotype was targeted using the combination of anti-KIR2DL1/2/3/S2 (8C11) and anti-KIR2DL2/3/S1/S2 (1A6) mAbs. Scatter plots showing ex vivo potential of KIR2DL1^+^/2/3/S1/S2^−^NK cell degranulation (CD107a^+^) against the HLA class I negative 721.221 B-EBV cell line (E/T ratio,10:1) in (**b**) C2^-^ (filled circles, *n* = 16) vs. C2^+^ (clear circles, *n* = 33) and (**c**) in C2^+^ individuals depending on allotypes (L1*003, *n* = 12, L1*002, *n* = 7 and L1*004, *n* = 15). (**d**) Density plots illustrating the targeting strategy of KIR2DL2/3^+^ KIR2DL1/S1/2^−^ NK (CD3^−^ CD56^+^) cells. A first combination of KIR2DL2/3/S2 (GL183) and KIR2DL1/S1 (EB6) specific mAbs leads to target KIR2DL2/3/S2^+^ KIR2DL1/S1^−^ NK cells. A second combination using anti-KIR2DL3 and anti-KIR2DL3/S2 (1F12) leads to target KIR2DL2^+^ KIR2DL3^−^/S2^−^ and KIR2DL3^+^ NK cell subsets. Scatter plots showing ex vivo potential of KIR2DL2/3^+^ KIR2DL1/S1/2^−^ NK cell degranulation (CD107a^+^) against the HLA class I negative 721.221 B-EBV cell line (E/T ratio, 10:1) in (**e**) C1^+^ individuals with cenAA (filled black circles, *n* = 15) or cenAB (clear circles, *n* = 15) motifs and (**f**) depending on allotypes (L2*001, *n* = 8, L2*003, *n* = 8, L3*001, *n* = 10, L3*002, *n* = 3 and L3*005, *n* = 2). Statistical differences were analyzed using unpaired t-test or one-way ANOVA followed by Turkey’s multiple comparison test. * *p* < 0.05, ** *p* < 0.01, **** *p* ≤ 0.0001.

**Figure 6 cancers-12-03595-f006:**
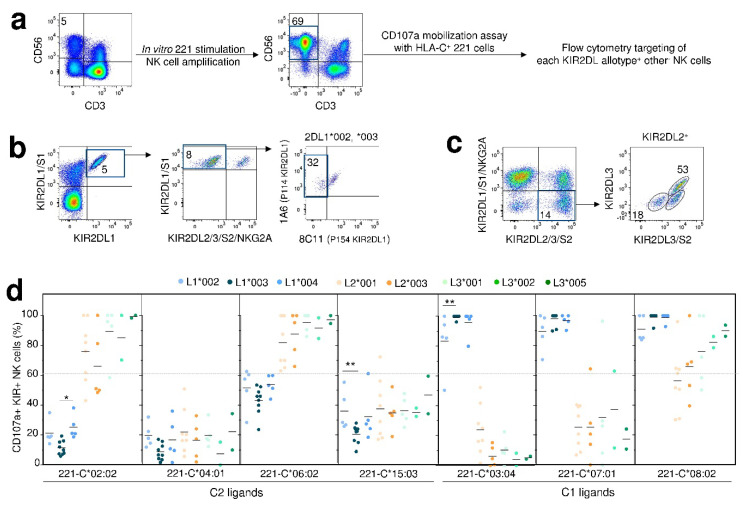
Broad disparity of HLA-C recognition between studied KIR2DL allotypes. (**a**) Strategy to analyze the degranulation potential of pre-stimulated KIR2D^+^ NK cells, against a panel of HLA-Cw transfected 721.221 cell lines. Density plots illustrating the amplification of NK cells (CD3^−^ CD56^+^) after 221 stimulation. The KIR2DL^+^ NK cell degranulation was assessed against 221 target cell (negative control) and different HLA-Cw transfected 221 target cells. (**b**) Density plots illustrating the strategy to target KIR2DL1^+/^2^−^/3^−^/S1^−^/S2^−^/NKG2A^−^ (KIR2DL1^+^ others^−^) NK cells and focus on each KIR2DL1 allotype using the combination of anti-KIR2DL1/2/3/S2 (8C11) and anti-KIR2DL2/3/S1/S2 (1A6) mAbs. (**c**) Density plots illustrating the strategy to target KIR2DL2^+^ and KIR2DL3^+^ NK cells from KIR2DL2/3^+^/L1^−^/S1^−^/S2^−^/NKG2A^-^ (KIR2DL3^+^ others^−^) using the combination of anti-KIR2DL3/S2 (1F12) and anti-KIR2DL3 mAbs. (**d**) Scatter plots showing the degranulation potential (CD107a^+^) of KIR2DL1+ other^−^ (blue circles) KIR2DL2^+^ other^−^ (orange circles) and KIR2DL3+other^−^ (green circles) for each allotype (L1*002 *n* = 5, L1*003 *n* = 9, L1*004 *n* = 5, L2*001 *n* = 8, L2*003 *n* = 5, L3*001 *n* = 5, L3*002 *n* = 2 and L3*005 *n* = 2). The mean ± SD is indicated. Statistical differences were analyzed using unpaired t-test or one-way ANOVA followed by Turkey’s multiple comparison test. * *p* < 0.05, ** *p* < 0.01.

**Figure 7 cancers-12-03595-f007:**
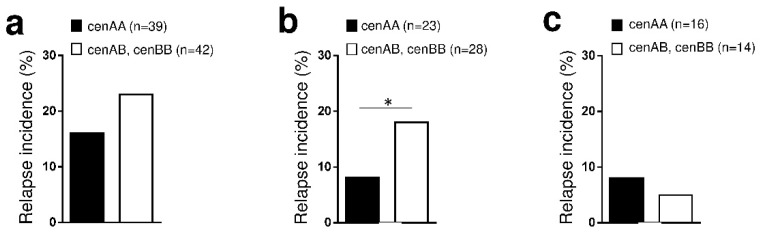
Beneficial impact of cenAA HSC donors on relapse incidence after T replete haplo-identical Hematopoietic Stem Cell Transplantation (HSCT) in myeloid diseases. Relapse incidence (%) observed after T-replete haplo-identical HSCT with post-transplant cyclophosphamide depending on donor centromeric KIR motifs (cenAA vs. AB/BB) in (**a**) all patients (*n* = 81), (**b**) restricted to patients with myeloid diseases (*n* = 39) and (**c**) restricted to patients with lymphoid diseases (*n* = 42). CenAA donors are KIR2DL2^−^/S2^−^/L3^+^, cenAB donors are KIR2DL2^+^/S2^+^/L3^+^ and cenBB donors are KIR2DL2^+^/S2^+^/L3^−^. Statistical differences were analyzed using unpaired t-test. * *p* < 0.05.

**Table 1 cancers-12-03595-t001:** Multivariate analysis of variables affecting relapse after T-replete haploidentical HSCT using post-transplant cyclophosphamide in patients with myeloid diseases.

Variables	HR [95CI%] ^1^	*p* Values
DRI high/very high vs. intermediate ^2^	5.38 [1.99–14.5]	0.0009
Donor HSC CenAA motif (positive vs. negative) ^3^	0.42 [0.18–0.99]	0.048

^1^ HR, hazard ratio; CI, confidence interval; ^2^ DRI, Disease risk index; ^3^ HSC hematopoietic stem cell; CenAA, centromeric KIR AA motif.

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
