# Peer review of "Centromeric KIR AA Individuals Harbor Particular KIR Alleles Conferring Beneficial NK Cell Features with Implications in Haplo-Identical Hematopoietic Stem Cell Transplantation"

_cancers, 2020, doi:10.3390/cancers12123595_

Round 1
Reviewer 1 Report
The donor deeply investigated KIR2DL NK cell repertoire in combining high-resolution KIR allele typing and 26 multicolor flow cytometry from a cohort of 108 blood donors with interesting in-vitro findings. Nervertheless, the trasposition of these results in a in-vivo setting is not supported by appropriate analysis, therefore title and conclusions are misleading. Univariate and multivariate analysis for relapse incidence are necessary to assert that cenAA HSC grafts limit significantly the incidence of relapse in patients with myeloid diseases after T-replete haplo-identical HSCT.
Reviewer 2 Report
I think this work is an excellent example about the relevance of KIR allelic polymorphism could have in the many association studies carried out in different phenotypes to date, including various types of cancer. In any case, I have some points that they would need clarification:
- Graphical abstract does not look too good, even there are some things that are not made clear (arrow + percentage on the left side, for example).
- The structure of KIR genes, their location, variability in gene composition, organization in different haplotypes, and now the influence of KIR polymorphism, are concepts that can be difficult to understand by the reader. Do you believe it is possible to design a figure to clarify this part (lines 64-90)?
- >80% of the results are possible thanks to the application of NGS in KIR allele resolution. Although it is with a reference, I find this matter is comparatively undeveloped in Materials “KIR2DL1/2/3/S1/S2 alleles were assigned on all healthy individuals by NGS [31].”
- Although it is specified along the discussion, twice at least, what worries me most is the possible influence of peptides in the results/conclusions. I would like to see more discussion into this matter (discussion + references). In addition, I miss a recognition of limitations of the study, by the authors.
- Minors.
Typographical error: line 164 (Figure 2b). ).
The title is too long, I think. Is it possible to consider a restructured?
In this Review, the authors describe the state of play in genetic testing for the purposes of exercise prescription and injury prevention. It is clear that the rapid development of genetic and genomic techniques has led to an increase in interest in the genetics of physical activity and sport. The manuscript is presented in appropriate manner, its scope is clear and very interesting.
Challenges for genetic testing in sport and exercise medicine are listed clearly and well. There are only two minor issues that should be taken into consideration:
- Cohort homogeneity could be further discussed. That is a real problem of translation. It looks like the development of evaluation tools will benefit from individual genomic information, increasing the predictive value and enabling the translation of the latest genetic testing results into clinical practice, but not all of these applications may be extrapolated from the Caucasian experience into Latin American populations, for example.
- Gene-by-sex interactions in exercise/injury genomics. Compelling findings of sex-dependent genetic effects on disease have been made (adiposity-related anthropometric traits, T2 diabetes, IBD). Is a different test necessary for males/females?, for example. Please, discuss it.
Round 2
Reviewer 1 Report
I suggest riconsidering the title and the strong message you give about reduced incidence of AML relapse in patients harboring centromeric KIR AA
Author Response
I suggest riconsidering the title and the strong message you give about reduced
Response: We agree with the reviewer’s comment. Thus we modified the title to focus mainly on the in vitro results and we removed “to limit myeloid leukemia relapse after”. The new title is “Centromeric KIR AA individuals harbor particular KIR alleles conferring beneficial NK cell features with implications in haplo-identical hematopoietic stem cell transplantation”. Moreover, in the conclusion section, line 520, we added “Nevertheless, our conclusions have to be taken with caution and to be confirmed from a larger cohort of haplo-identical HSCT donor/recipient pairs.”

Reviewer 2 Report
Congratulations, very good work.
Author Response
We thank the reviewer 2 for his positive comment.
Round 3
Reviewer 1 Report
I appreciate both title and text editing at line 520.